# Analysis of the Accuracy Potential of a Stereo High-Speed Camera System in 3D Measurements in Highly Dynamic Experiments

**DOI:** 10.3390/s23042158

**Published:** 2023-02-14

**Authors:** Laura Camila Duran Vergara, Frank Liebold, Hans-Gerd Maas

**Affiliations:** Institute of Photogrammetry and Remote Sensing, TU Dresden, 01062 Dresden, Germany

**Keywords:** high-speed cameras, dynamic 3D measurements, accuracy, rigid-body transformation

## Abstract

In the context of setting up a stereo high-speed camera system for accurate 3D measurements in highly dynamic experiments, the potential of a *“Fastcam SA-X2”* stereo system is evaluated by testing different camera configurations and motion scenarios. A thorough accuracy analysis is performed using spatial rigid-body transformations and relative measurement analyses of photogrammetrically reconstructed surfaces of nondeformable objects. The effects of camera calibration, exposure time, object velocity, and object surface pattern quality on the quality of adjusted 3D coordinates are taken into consideration. While the exposure time does not significantly influence the quality of the static measurements, the results of dynamic experiments demonstrate that not only an insufficient frame rate but also an increased noise level resulting from short exposure times affects 3D coordinate accuracy. Using appropriate configurations to capture dynamic events, the errors in dynamic experiments do not differ significantly from the errors obtained in static measurements. A spatial mapping error of less than 1 μm is obtained through the experiments, with proper testing configurations for an object surface area of 5×20 mm. These findings are relevant for users of high-speed stereo imaging techniques to perform geometric 3D measurements, deformation, and crack analyses.

## 1. Introduction

### 1.1. General Background

Due to their high spatio-temporal resolution, high-speed optical systems are widely used to quantify the behavior of objects in dynamic events. A compilation of structural dynamics applications using photogrammetric techniques can be found in [1]. The benefits of optical devices and other measurement techniques are compared here, emphasizing the advantage of noncontact as a practical aspect of photogrammetric measurements. Accurate measurements can be obtained by using stochastic patterns applied to the object surface, as random speckle information can be uniquely recognized in the recorded images. Image matching is used to track the pattern field information and generate surface points at each time step of the recorded event.

The result is a continuous full-field formed by surface meshes with spatial coordinates of the generated points. These coordinates are the basis for the evaluation of the structural behavior via different mechanical analyses such as deformation [2,3,4,5], crack propagation [6,7], wave propagation, etc.

High strain rates or impact velocities are applied to different structures to assess the response of the material to the dynamic tests. Because of the applied high strain rate, the time of interest for analyzing the material deformation in these impact experiments is short, and high frequency measurements are needed. In addition, the stepwise relative deformation of the material during this time of interest may lie in the micrometer range for small-scale objects. For this reason, the mechanical properties derived from the optical measurements can lead to false conclusions if the quality of the photogrammetric measurements is insufficient.

While capturing short-time dynamic events, image blur is an obvious motion effect that increases the uncertainty of image matching processes. An uncertainty in dynamic conditions is explored in [8] by comparing the reference measurements from accelerometers and laser sensors with optical measurements in controlled vibration tests. Different motion scenarios as well as acquisition parameters are also tested in [9], where the blurring effect is also quantified by some indices using reference laser measurements and Spatial Frequency Response Analysis. Simulations of the motion effect are presented in [10] to estimate the uncertainty of Digital Image Correlation (DIC) and they are validated in real dynamic tests by imposing harmonic motion to a target. In addition to validating and simulating this motion effect, a numerical technique is proposed in [11] to predict the motion effect and to thus predict the DIC uncertainty to improve the performance of dynamic applications. Besides this uncertainty estimation in rigid targets, accuracy analyses in deformable objects are evaluated in [12], where the effects of camera settings and motion parameters on DIC methods are investigated and validated with laser sensor measurements. In [13], there are also different methods that have been proposed to compensate for the amount of blur varying in a single frame and to consequently improve the accuracy of the optical measurements.

Image blur can be minimized by decreasing the exposure time in the camera configuration. Reducing the exposure time requires an increase in illumination to ensure an optimal signal-to-noise ratio (SNR) in the images. As explained in [14], the photon noise arises from the statistical noise of the number of photons hitting the sensor. This noise depends not only on the light sensitivity and fill factor of the sensor but also on the ratio of signal to integration time.

Despite the drawbacks presented above, the field of civil engineering often resorts to these systems to monitor concrete structures loaded with different forces and impact directions. As an example, the impact resistance of existing building materials is analyzed by the Research Training Group GRK 2250 of the German Research Foundation [15]. In the scope of this group, the use of optical systems is often favored because of their ability to provide time-resolved precise areawise noncontact 2D or 3D measurements. Image sequence evaluation for surface reconstruction and material analysis is carried out using the software “GOM Aramis”.

### 1.2. Research Problem Analysis

Appropriate configurations of the optical systems for accurate measurements in highly dynamic experiments are needed in order to correctly describe the performance of the tested materials within the research group GRK 2250. As mentioned before, the motion blur resulting from recording moving objects with a long exposure time affects the image matching process for coordinate determination. Moreover, a short exposure time can deteriorate SNR, degrading the image quality and therefore the measurement accuracy.

As stated before, external devices have been used to quantify the uncertainties coming from photogrammetric measurements. A consistent external validation of optical measurements with other devices such as accelerometers, strain gauges, or pointers is difficult to provide for high frequency spatially resolved experiments. The accuracy of these sensors depends on the frequency; therefore, it varies. The accuracy given by the manufacturer often only applies to lower frequencies than the frequencies used by high-speed camera systems. Furthermore, the sensors are often not able to capture a surface but only discrete points. The measurement direction of these sensors is in most cases uniaxial, and its exact orientation within the orientation of the optical device, either in the *X*, *Y*, or *Z* axes, is complex to ensure. Furthermore, the same position over the object surface could not be evaluated because the sensors would restrain the camera field of view.

Alternatively, the potential of photogrammetric measurements can be assessed using the internal precision statements of the adjusted parameters for camera calibration and coordinate determination. However, these statements needed for an accuracy analysis of the results are not provided by “GOM Aramis”.

### 1.3. Structure

To overcome these limitations, the accuracy analysis methodology for 3D high frequency testing proposed in this work is based on spatial similarity transformations, together with 3D relative measurements. Both analyses are based on the surface point coordinates determined by “GOM Aramis”. The transformation analysis is limited to object measurements without deformations, because it is not possible to make assumptions from a spatial similarity analysis regarding the deformation behavior. Since there is no deformation in the measurements, the scale factor is fixed for all transformations and set to 1. This special case of 3D similarity transformations is called rigid-body transformation (RBT). The basic idea of the rigid-body motion evaluations is the analysis of the spatial mapping error variations, which reflect the abovementioned degradation effects.

The proposed analysis is performed for two event conditions: First, a static object was recorded at different frame rates to analyze the effect of short exposed images on the temporal and geometrical stability of points surfaces without motion. Furthermore, the effect of different exposure times when capturing images for camera calibration on the 3D coordinate determination is also evaluated. Second, a moving object was recorded with the same scheme of exposure times, as was used in the static case. Here, in addition to the geometrical stability of the reconstructed surfaces, the dependency of accuracy on parameters such as camera exposure time and object velocity are analyzed.

This work is structured as follows: First, the experimental setup for static and dynamic measurements is presented. Afterward, the methodology for accuracy analysis based on RBT and different quality parameters is presented. The results of the transformations and 3D relative measurements, as well as image performance analyses, are then presented. Finally, the possible reasons for the results are discussed and concluded. The complete experimental data and results are available upon request from the author responsible for the correspondence.

## 2. Experimental Setup

Systematically generated patterns (SPs) [16] and randomly generated patterns (RPs) as presented in Figure 1 were captured with a stereo high-speed camera system for 3D measurements, consisting of two *“Photron Fastcam SA-X2”* cameras. The main technological specifications of this camera are listed in Table 1. Besides evaluating various stochastic patterns with a surface area of 100 mm${}^{2}$, testing configurations such as camera calibrations, object velocities, and exposure times were taken into account under static and dynamic event conditions. The calibrations were performed at the same location as the static pattern field and the dynamic pattern field entered the cameras’ field of view (FOV) when dropped. In contrast to the full sensor resolution used for the 3D calibrations, the cameras’ FOV used for the measurements was 256 × 336 px (pixel). This resolution was selected in order to achieve the highest possible configurable frame rate that can capture the entire width of the surface. While the cameras were manually triggered for calibration and static recording, for dynamic recording the triggering occurred when the first pattern fully entered the camera’s field of view with the help of a “Through-beam sensor”. An appropriate illumination for short exposure times and homogeneous light distribution was ensured using two *“1kW-LED-4438”* lamps in continuous mode, placed on the left and right sides of the camera system. The evaluation of the images in order to obtain the 3D coordinates, and the corresponding camera calibration, were carried out using the software “GOM Aramis”.

The evaluated static and dynamic testing configurations are presented in the workflow of Figure 2. For the 3D camera calibration process, every set of calibration images was recorded with 5 different exposure times. In order to obtain the same calibrated volume geometry, every calibration plate position was captured with the 5 exposure times before changing to the next calibration plate position, as presented in Figure 3. The 05 camera calibrations were used in order to check their effect on 3D coordinate determination using the statically recorded pattern surfaces. These pattern surfaces were recorded with 7 exposure times in the same frame time. The latter aims for observing a possible camera behavior independent of the exposure time. On average, each pattern surface consists of 50×200 pixel (px). The exposure times were defined by selecting the longest possible duration of exposure by setting the camera configuration to the 7 different frame rates. Good lighting for the images was ensured by checking beforehand that there is no low contrast, i.e., gray values of light areas ≥150, or overexposure, i.e., gray values of light areas <255.

The dependency of the accuracy on the dynamic event conditions and camera configurations was evaluated using a single 3D Camera Calibration, as presented in Figure 2. The tree velocities were obtained by dropping the drop-weight from 3 different heights. These dynamic events were recorded with 7 frame rates (or exposure times), but depending on the drop-weight velocity, the testing configurations 1000 fps and 5000 fps are not taken into account for the analyses. Capturing an object at a high velocity with a long exposure time generates extremely blurred images, as shown for example, in Figure 4. Low contrasted images offer insufficient information for an appropriate matching process and thus impair the determination of the surface points’ positions.

From the presented testing configurations, 770 and 352 experimental configurations (presented in Table 2) were tested; therefore, the same number of data sets was obtained.

## 3. Methodology

To qualify the potential of high frequency optical measurements, the following uncertainty sources were taken into account during testing:(i)Tested:Pattern quality: Different levels of stochastic information are provided in SPs and RPs.Exposure time: Its effect on image capturing is evaluated in static and dynamic event conditions.Frame rate: It is also included to the extent, as it limits the maximal exposure time.Camera calibration: Test field camera calibrations were performed using the calibration plate *“CP20 90 mm × 72 mm”* produced by the software provider. A test field calibration is characterized by the fact that reference points and reference lengths are known [18]. In this case, two reference distances given by the calibration plate are identified by the software from the calibration images to perform the camera calibration process.(ii)Considered:Sensor cleaning: Before testing, the cameras sensors were cleaned to avoid errors coming from dust particles.Lighting: Adequate brightness and contrast before and after recording were verified for all captured images.*Facet size* and *point distance* for DIC: The facets are square sections in the right and left images. The point distance describes the distance between the center points of the adjacent facets. For all the experiments, a *facet size* of 19 px and a *point distance* of 16 px were set.*Slider distance* and *measuring distance*: These denominations are defined by the software, respectively, as the distance between the camera sliders and the distance between the object and the bar on which the cameras are held. This is different from the stereo baseline and the object distance, which are defined in photogrammetry as the distance between the perspective centers of the cameras and the distance between the baseline and the object. For the calibration plate and a camera lens of 75 mm, a *slider distance* of 105 mm and *measuring distance* of 620 mm is recommended by the software. These correspond approximately to a baseline of 200 mm and an object distance of 470 mm.(iii)Negligible:The variation of the drop-weight velocity obtained from the same height due to friction problems in the facility.External conditions such as temperature variation and vibrations, since the tests were performed in a controlled lab environment. A lab temperature of 20 ${}^{{}^{\circ}}$C was set for the camera calibrations.

The workflow shown in Figure 5, considering the aforementioned uncertainty sources, is followed to evaluate the accuracy potential of high frequency measurements recorded with the stereo high-speed camera presented. First, the “Static Testing Configurations” are employed to evaluate the significance of the “3D Camera Calibrations” on the 3D coordinate determination using RBTs. Due to the insignificant influence of the exposure time on the camera calibration for the surface determination (presented in Section 4.1), only one “3D Camera Calibration” is used for the coordinate determination of the “Dynamic Testing Configurations”. Based on the RBT results, static and dynamic experimental configurations were selected for relative 3D measurement analysis. On the one hand, the data sets of the static measurements of the best performing RP and SP surfaces are used for temporal measurement stability. These data sets were obtained by using one of the tested calibrations. On the other hand, the data sets of the best dynamic RP and SP measurements with the highest velocity are used for geometrical measurement stability. To have comparable conditions, the geometrical stability of the data sets from the static measurements obtained with the same frame rate is analyzed.

The internal quality variables *intersection deviation* and *stereo residual* given by “GOM Aramis” are also taken into account to evaluate the coordinate precision given by the software itself. The *intersection deviation* describes how much the calculated 3D coordinate position deviates from the calculated 3D coordinate of the same point based on the calibration. The *stereo residual* describes the difference in the gray value distribution of a facet between the left and right camera and is specified in gray values.

Since these internal quality variables did not provide enough information to explain the quality of the experimental configurations, a performance evaluation of the surface pattern images was carried out to support the results of the RBTs and relative 3D measurements. From the recorded static images, the best performing RP and SP surfaces were cropped using the corners’ pixel positions. To obtain the pattern surface from the dynamic recordings, the pixel accuracy method *Template Matching Correlation Coefficient* presented below is used to match the pattern surfaces. Here, the pattern surface (*P*) with pixel coordinate position (x’,y’) is searched in the image (*I*) by locating the resulted pixel position R(x,y) with the highest matching probability.
(1)$$R\left(x,y\right)=\sum_{x^{\prime },y^{\prime }}\left(P^{\prime }\left(x^{\prime },y^{\prime }\right)\cdot I^{\prime }\left(x+x^{\prime },y^{\prime }+y^{\prime }\right)\right)$$

The cropping and the matching process are applied to the images captured with the right and left camera of the corresponding experimental configurations. The parameters brightness ($B_{P}$), standard deviation ($Std_{P}$), and contrast ($C_{P}$) defined in [19] per pattern (*P*) image, as well as the SNR per pattern image pixel ($P_{px}$) based on [14], are computed for the resulting images, and they are defined as follows:(2)$$B_{P}=\frac{1}{m\times n}\sum_{x=0}^{m-1}\sum_{y=0}^{n-1}P\left(x,y\right)$$
(3)$$C_{P}=\frac{g_{max}-g_{min}}{g_{max}+g_{min}}$$
(4)$$Std_{P}=\sqrt{\frac{1}{m\times n}\sum_{x=0}^{m-1}\sum_{y=0}^{n-1}{\left[P\left(x,y\right)-B_{P}\right]}^{2}}$$
(5)$$SNR_{P_{px}}=\frac{B_{P_{px}}}{Std_{P_{px}}}$$
(6)$$B_{P_{px}}=\frac{1}{T}\sum_{t=0}^{T-1}P_{px_{t}}\left(x,y\right)$$
(7)$$Std_{P_{px}}=\sqrt{\frac{1}{T}\sum_{t=0}^{T-1}{\left[P_{px_{t}}\left(x,y\right)-B_{P_{px}}\right]}^{2}}$$
m×n is the total number of pixels (px) a pattern consists of, with the pixel coordinate (x,y) positions being formed by (*m*) rows, (*n*) columns, and (*g*) gray values. The $B_{P_{px}}$ and $Std_{P_{px}}$ are used to calculate the $SNR_{P_{px}}$ iterating over every time step (*t*) of the total (*T*) images captured by the corresponding experimental configuration.

With these quality parameters, a descriptive statistical analysis is carried out. In order to obtain the most typical value from the different experimental conditions, the median was selected to represent the transformation experimental results and the internal quality variables of “GOM Aramis”.

### 3.1. Rigid-Body Transformation

This spatial rigid-body transformation is used to define the spatial mapping of shape-invariant objects located at two different time steps *t* in the same coordinate system (X, Y, Z), as presented in Figure 6. These object locations can be arbitrarily rotated, shifted, and scaled with respect to each other. Since no deformations are applied, the scale parameter is set as being equal to 1. Therefore, this RBT is adjusted using 6 parameters: 3 translations (TX, TY, TZ) and 3 rotation angles (ω, ϕ, κ). Although the parameter estimation basically only requires 6 observations, the system of equations will usually be overdetermined for the sake of redundancy. An optimal solution of the equations is then achieved by using a least-squares adjustment [19].

For each experimental configuration, a data set is obtained and formed by *n* point clouds with ($TX_{n}$, $TY_{n}$, $TZ_{n}$) coordinate points. The 3D coordinate determination and export of the *n* point clouds are carried out in “GOM Aramis” to perform (n−1) RBTs, as explained in Figure 5. As an output, if there are no systematic errors in the measurements, the statistical value of the standard deviation of unit weight $s_{0}$ is used to qualify the precision of each RBT adjustment. This quality parameter indicates the precision of the deviation between the transformed coordinates. The $s_{0}$ deviation is obtained from the observation residuals and the redundancy of the equation system. In this work, the median of $s_{0}$ resulting from the (n−1) RBTs is denominated as spatial mapping error.

The evaluation of RBT is used in this work to determine the influence of different experimental configurations on the determination of the surface coordinates calculated in “GOM Aramis”. It is obviously related to the relative (rather than absolute) coordinates and accuracies, as those are of central relevance in dynamic measurements using stereo high-speed camera systems.

### 3.2. Relative Measurements Analysis

As schematized in Figure 5, static and dynamic data sets are used to evaluate the relative measurement accuracy of the best experimental configurations. For the temporal stability analysis, the average of the displacements of all surface points per time step are calculated for the pattern surfaces that are not in motion. In this publication, these motionless measurements are called total apparent displacements (DXYZ) and are defined in Formula (Equation 8). This evaluation is considered for the 7 exposure times. The averaged apparent displacement for axis direction such as DX, DY, and DZ are also considered. Random errors, that are not constant or reproducible in the measurements are the cause for these pseudodisplacements.
(8)$$DXYZ=\sqrt{DX^{2}+DY^{2}+DZ^{2}}$$

The geometrical stability analysis is performed by evaluating the effect of random errors on constructed geometries within a pattern surface mesh. Temporal length variation (ΔL) of a line formed by two coordinate points is observed for static and dynamic measurements. Figure 7 presents the ΔL between two points, as well as the variation per axis direction, ΔLX, ΔLY, and ΔLZ.

Because the external conditions are considered negligible in the performed experiments, these apparent displacements and temporal length variations can be used to analyze the stability of the camera system.

## 4. Experimental Results

### 4.1. Rigid-Body Transformations

The performed transformations of each static experimental configuration are summarized using the median of the empirical standard deviations $s_{0}$. These spatial mapping errors, obtained by using one calibration, are presented in Figure 8 for the RPs and SPs, since the applied calibrations and recorded frame rates result in a similar response for both types of generated patterns (refer to Section A.1 and Section A.2). This means that the tested camera exposure times for capturing the calibration images lead to a consistent determination of the calibration parameters under appropriate lighting conditions. For this reason, the adjusted systematic errors of each camera calibration have no significant influence on the determination of the 3D space coordinates; thus, the spatial mapping error remains almost unchanged for the tested static experimental configurations. Contrary to this, the influence of the pattern quality of the listed patterns in Figure 2 is clearly reflected in the results. From the RPs in Figure 8A, the highest performance is given by 05 and 06 and the lowest by 01 and 11. From the SPs in Figure 8B, 02 and 08 show the highest performance, and 05 and 06 show the lowest.

The spatial mapping errors of both types of generated patterns demonstrate that the amount of stochastic information influences the results, since in general the RPs show lower deviations. However, in addition to stochastic information, a good contrast is also needed for the quality of the captured signal to be adequate for the reconstruction and mapping process of the speckle surface. This is the case of the SP 02.

Due to the fact that the image exposure time does not significantly influence the camera calibration and thus the 3D coordinate accuracy, a single camera calibration is used for the dynamic measurements. The spatial mapping errors for the dynamic experimental configurations are presented in Figure 9 and Figure 10. It can be seen that the frame rate affects the SPs and RPs differently. While the SP results are influenced more by the image noise than by blurring, the RP results of the 01 and 02 Height are influenced more by blurring. This implies that a good contrast allows better matching results even if there is blurring. However, the 03 Height RP results show a larger effect of image noise than that of blurring.

Comparing the spatial mapping error results of the experiments with and without motion, the error differences between the static and dynamic measurements captured with the same camera configurations are higher with a falling pattern quality. In Table 3, the median of the $s_{0}$ values for the best- and worst-performing patterns are presented. Taking the low-performing patterns 05 and 01, the error of the SP is significantly higher due to the lack of stochastic information of the printed surface. Comparing the high-performing patterns, the error of SP 02 delivers a slightly lower s0 than RP. Due to the difficulty of generating a consistent and proper RP, it can be assumed that in general, a SP has the potential to enhance the quality of the measurements. This gains more relevance for small-scale objects and specific experiments that are intended to be repeated several times.

Using the computed point coordinates corresponding to the dynamic experimental configurations of the best- and worst-performing patterns, the following values are presented in Figure 11 and Figure 12 in order to explain their performances: Number of tracked points per time step, vertical cumulative object translation (TY), the average distance between the surface points per time step, and the recording time.

The performance of the worst patterns is the result of a fluctuating number of tracked points over time and a larger distance between them. The higher performance of SP 02 compared to RP 06 is due to the higher number of tracked points and the associated smaller distance between them. When analyzing the behavior of the number of tracked points and the average distance between them by experimental configuration, it can be noted that in general the results are noisier with higher drop-weight velocity and image frequency. Furthermore, the number of points fluctuates more than the distance between them. This means that the surface geometry obtained from an experiment is not affected by the frame rate used.

In addition, the $s_{0}$ errors of the dynamic transformations show that an increasing frame rate enhances the results, but at an even higher frame rate, the quality of the results can decrease again. From the lowest $s_{0}$ values marked in bold in Table 3, it can be concluded that the object velocity does not significantly influence the coordinate determination, if the exposure time is appropriate. Although the differences between the static and the dynamic bolded values are slightly larger for the RP, the differences are not more than half a μm for suitable experimental configurations.

A dependence between object velocity and camera exposure time can be noticed in the results of the dynamical experimental configurations. This means that the higher the velocity, the better the performance at a higher frame rate. This is most evident in the results using the RP 06, where for the first, second, and third velocities, the frame rates 10,000 fps, 25,000 fps, and 50,000 fps perform the best, respectively. Figure 13 and Figure 14 illustrate this behavior more clearly with the obtained $s_{0}$ from all the transformations for the best-performing patterns.

The influence of exposure time is illustrated in the boxplot distribution of the resulting $s_{0}$ values. On the one hand, a larger dispersion of the standard deviations is achieved through long exposure times, which lead to low contrasted images because blurred images present similar gray values of neighboring pixels due to the motion. On the other hand, the resulting $s_{0}$ spread at high image frequencies is caused by image noise, since the probability that the light-sensitive part of the pixels detects an object steadily over time decreases with decreasing exposure time. However, the s0 distribution of these experimental configurations shows that performance is more affected by image noise than by blur effects. The effect of image noise can be seen in Figure 15 and Figure 16 for the patterns with the best performance under the third dynamic condition, where the $SNR_{P_{px}}$ decreases with decreasing exposure time for the tested frame rates. This main tendency is also observed for the SP 02, although the SNR is higher compared to the RP 06.

An attempt is made to find the source of the higher errors with increasing frame rate using the quality dimensions *stereo residual* and *intersection deviation* given by “GOM Aramis”. No direct correlation is found between the $s_{0}$ values and the aforementioned dimensions. It can only be seen in Figure 17A that the gray value difference between right and left image facets increases with raising frame rate, possibly affecting the image matching performance. The *intersection deviation* in Figure 17B also does not show an influence on the precision of the transformed coordinates but only variable discrepancy between the observation rays calculated from the measurements and the calibration. These discrepancies are the result of random errors that do not show a correlation with the tested exposure times. Similarly, the end velocity computed by the software at different frame rates in Figure 17C does not seem to be correlated with the $s_{0}$. The measurement stability of the reconstructed geometry for all tested frame rates, shown in Figure 12, may be the reason for this.

### 4.2. Relative 3D Measurements

The transformation results show how the testing configurations can affect the spatial mapping of a rigid-body during the measurement under different event conditions. However, in addition to surface reconstruction and surface point tracking, the measurements derived from the surface points coordinates are also of interest for relative accuracy analysis. The computed displacement by “GOM Aramis” is first evaluated in static recordings in order to observe the variation of the point coordinate position over time, which varies because of random external conditions. Then, the length variation between two points is considered to evaluate the stability of relative measurements in both event conditions, static and dynamic.

As presented in Figure 5, the relative 3D measurement analysis is carried out for the experiments in static condition using the 01 Calibration and in dynamic condition using the 03 height for the best-performing patterns RP 06 and SP 02. The evaluation of image performance parameters for the previously mentioned experimental configurations is used to support the results. Therefore, $B_{P}$, $C_{P}$, and $Std_{P}$, defined in Formula (Equation 2)–(Equation 4) are presented in Appendix B.

#### 4.2.1. Temporal Measurement Stability

The total apparent displacements DXYZ defined in Section 3.2, Formula (Equation 8), are presented for the seven recorded frame rates in Figure 18 for RP 06 and SP 02. The range variation of the displacements behaves differently for each tested frame rate, as can also be noticed in Table 4 and Table 5. However, there is a trend of increasing variation of the apparent displacement with decreasing camera exposure time. It can be assumed from Section B.1 and Section B.2 that the image $B_{P}$ level does not influence the apparent displacement, while the latter may be affected by the difference of $C_{P}$ and $Std_{P}$ between the imaged pattern captured with the right and left cameras. For example, RP 06 and SP 02 recorded at the frame rates 50,000 fps and 10,000 fps, respectively, have the smallest differences in $C_{P}$ and $Std_{P}$, and the smallest variation in apparent displacement. Furthermore, the shape curve of $Std_{P}$ behaves similarly to the shape curve of the apparent displacement. This is especially evident in the measurements taken at the highest frame rate. The temporal measurement stability is thus influenced by the gray level differences in the images for the matching process, short exposure times, and image noise. A larger range variation of the RP 06 apparent displacements is caused by the lower contrast quality presented in the image performance analysis.

A systematic oscillation in Figure 18 can also be found in the time component. These oscillations are also present in the apparent displacements per axis direction, but the oscillation in the *Z*-axis contributes more to the amplitude of the total apparent displacement, as shown in the range variations in Table 4 and Table 5. This is due to the convergence geometry of the stereo camera system. The frequency period of the apparent displacement occurs approximately every 7.5 ms for all of the tested frame rates. Since the image performance parameters do not show any frequencies over the recording time, and there was no systematic external event during the experiments, the cause for these periodic oscillations may come from the camera’s internal operation mode. One possible cause of these frequencies is the constant vibration of the ventilation system of the camera. The fans can be turned off during the recording, but this action will make the measurements inconsistent as the camera sensor heats up and the fans slowly turn off.

#### 4.2.2. Geometrical Measurement Stability

The length variation ΔL between two points over time computed by “GOM Aramis” is used to analyze the stability of relative measurements within the pattern surface. The results of the length variations ΔL, ΔLX, ΔLY, and ΔLZ, as well as the initial lengths and the corresponding range of variation, are presented in Figure 19 and Figure 20. Similar to the results presented in the previous section (Section 4.2.1), the values of ΔLZ in the Z-axis direction vary over a wider range than the values for ΔLX in the X-axis and ΔLY in the Y-axis, for both dynamic and static conditions. While in the static experiment the total ΔL variation is 2 μm for both patterns, there is a variation of 5 μm in the dynamic case when using the RP 06, as opposed to 2 μm when using the SP 02.

The total ΔL variation of the static measurements in Figure 19 present similar frequencies in the time component, as shown in the total apparent displacement in Figure 18, but the range of variation is 2 μm instead of 4.92μm for SP 02, and 2.26μm for RP 06. These systematic oscillations cannot be observed in the results of the dynamic tests in Figure 20, since the measurement time is less than 7.5 ms. A correlation with the image performance results is found, since the curve arching shape that appears in the length variation of the RP 06 is similar to the curve shape of the $B_{P}$ results with the same tested frame rate in Section B.3. The same arching shape does not appear in the $B_{P}$—50,000 fps curve in Section B.4 for SP 02, but it has a similar shape behavior with the length variation curve.

From these findings, it can be concluded that even small variations in light over time can affect the results. This is because the signal detected by the sensor pixels directly affects the quality of the matching process, and thus, the resulting 3D measurements. The matching errors or gray level differences for determining the different surface points vary within the image. This variation in $SNR_{P_{px}}$ is the result of short exposure times and can affect the final coordinate of each point differently. For this reason, the displacements of the points forming the line geometry differ from each other. To demonstrate this effect, the vector displacements of both points of a static test are presented in Figure 21. The higher contrast of SP 02 in comparison to RP 06 contributes to the increase in SNR, and consequently, to the smaller length range of variation achieved, which is particularly noticeable under dynamic conditions. Image noise can be observed in the pixel variations of the SP 02, especially in the light areas. The probability of obtaining less noise using randomly generated patterns is low, since speckle dispersion cannot be controlled.

## 5. Discussion

Different experimental configurations, varying camera settings, and motion parameters were systematically tested in order to evaluate the potential of the optical high-speed stereo camera system *“Fastcam SA-X2”*. Analyses of rigid-body transformations and relative coordinate accuracy were carried out on the basis of surface point coordinates and relative measurements computed by the software “GOM Aramis”. With the help of these studies, the accuracy determination of the 3D surface measurements of nondeformable objects was used to evaluate the stability of the stereo camera system.

A large number of high frequency experimental data were obtained to evaluate the influence of the experimental configurations in static and dynamic measurement conditions. For all of the tested experimental configurations, each pattern with a surface area of 5×20 mm was observed on average by 50×200 px. This means an observed surface pattern area per pixel of 100×100μm.

First, the spatial mapping errors ($s_{0}$) demonstrated that the exposure time does not affect the accuracy of stereo camera calibration; thus, neither the 3D space coordinate accuracies, if appropriate conditions, including lighting and camera setup configurations, are given. Unlike this, the quality of the applied pattern significantly influenced the surface reconstruction. The $s_{0}$ values of the highest-performing systematically and randomly generated patterns were 0.59μm and 0.67μm, in contrast to the lowest-performing patterns, respectively, with 5.83μm and 1.67μm. Although in general, the randomly generated patterns presented lower spatial mapping deviations, these deviations were even lower if an appropriate systematically generated pattern was used. This applied to the results performed under static and dynamic event conditions.

For the dynamic experiments, it was found that the $s_{0}$ values were slightly larger than the $s_{0}$ values obtained with the static measurements, if an appropriate exposure time was used to capture the dynamic events. The $s_{0}$ values of the dynamic conditions, compared to $s_{0}$ values of the static conditions, were not higher than 23 % for the best-performing patterns, as presented in Table 3. In addition, a dependence between object velocity and camera exposure time was found. This is due to fact that the $s_{0}$ values were lower when using a shorter exposure time for a higher velocity. Moreover, the performance of the measurements decreased at even shorter camera exposure times, as was the case with the results of the recordings captured at frame rates of 75,000 fps and 100,000 fps. The reason for this, despite the optimal image brightness, was the increasing pixel noise that reduces the pixel signal. Although the dynamic results were generally more affected by image noise, the blur effect degraded the quality of the images using the randomly generated patterns more than those using the systematically generated patterns. When using a high-quality pattern and an appropriate camera exposure time for the corresponding object velocity, the resulting spatial mapping errors did not exceed 1 μm. In this work, good experimental configurations for the three velocities tested reached an accuracy of 1/20 px based on the pixel size.

Moreover, the relative stability of the measurements derived from the surface point coordinates were evaluated. Image performance results were used to support the relative accuracy analysis. As a result of the convergence geometry of the stereo camera system, the temporal and geometrical measurement results varied in a larger range in the Z direction. In material analysis, the random errors per axis direction can affect the evaluated mechanical property in different ways. For the tested camera system, variations in the Z-axis direction will introduce larger errors for shear test analysis than for strain deformation analysis in the X- or Y-axis directions. In addition, a constant time period oscillation of approximately 7.5 ms was observed in the apparent displacements and lenght variations of static measurements. Due to the fact that the measurements were performed in a controlled environment and captured with a continuous illumination, these frequencies may come from the camera’s internal operation. The camera’s ventilation system or electronic components may be the reason for these vibrations. Since the time of interest of experiments under highly dynamic strain rates is usually short, these frequencies may be not visible, but they are present in the results. However, depending on the accuracy sought, these random errors must be taken into account.

The variation ranges of the length variations tested in this work were smaller compared to the variation ranges of the apparent displacements. The reason for this is that each determined point coordinate is affected differently due to random errors. These random errors include fluctuations in the pixel signal due to short exposure times. Therefore, the stability of the measurements may vary depending on the points used for the analysis. One possible solution to minimize the influence of random errors is to subtract the average of the apparent displacements of fixed points in each axis direction.

Finally, the higher contrast of the systematically generated pattern 02 compared to the randomly generated pattern 06 contributed to a lower influence of random errors. The result was a higher measurement stability of the nondeformed analyzed geometry.

## 6. Conclusions

Analyses of rigid-body transformations and relative object measurements were proposed to evaluate the potential of a stereo camera system. These studies allow for accuracy statements on photogrammetrically reconstructed surfaces to be made without deeper knowledge of the internal software processes for 3D coordinate determination. Proper experimental configurations for quantifying the object motion showed that not only a suitable exposure time can influence the performance of 3D measurements, but a good contrast of the pattern surface is also crucial to compensate for signal quality loss in high image frequency measurements. In suitable lighting conditions, it can be concluded that under dynamic conditions the blur effect of low contrast patterns compromises the matching process more than image noise.

Moreover, short exposure times can increase the differences in the gray values of the respective images for the matching process affecting the 3D surface reconstruction. It was also found that the temporal and geometrical stability of the relative measurements can be influenced by variations in the radiance of the object. As a result of considering the adequate optical system configurations depending on the object motion, slightly larger spatial mapping errors were obtained in the dynamic tests compared to the static results.

Cyclic apparent displacements of 3D coordinates have to be taken into consideration. While this is less relevant in experiments, where only relative measurements within the object surface are taken (e.g., crack width measurement), it may significantly degrade the quality of results of 3D object reconstruction, thus requiring additional effort. Errors introduced by these cyclic displacements can for instance be compensated by using static background targets in a suitable configuration.

Since the presented studies are based on spatial mapping analyses, rigid-body transformations can also be used to assess the accuracy of a single camera. In order to obtain the real dimensions of the object surface under study, a scaling factor must be used. The resulting 2D surface coordinates can also be evaluated under dynamic conditions, where the distance between the camera and the object should remain constant. Effects of the experimental setup used, such as the type of illumination, must also be taken into account in the analyses.

Rigid-body analyses to test the accuracy potential of a stereo high-speed camera system have the limitation that only a single velocity is considering during the experiment. Material deformations at high dynamic rates propagate at different velocities due to the material response of the structural behavior. This complicates the choice of an appropriate exposure time. However, a generalization of the experimental velocity can be used to set up the optical measurement system.

## Figures and Tables

**Figure 1 sensors-23-02158-f001:**
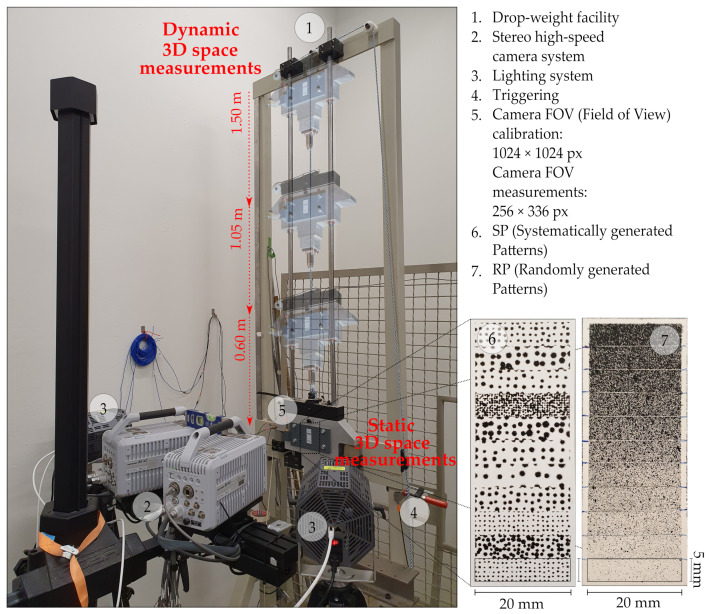
Measurement system configuration for static and dynamic event conditions.

**Figure 2 sensors-23-02158-f002:**
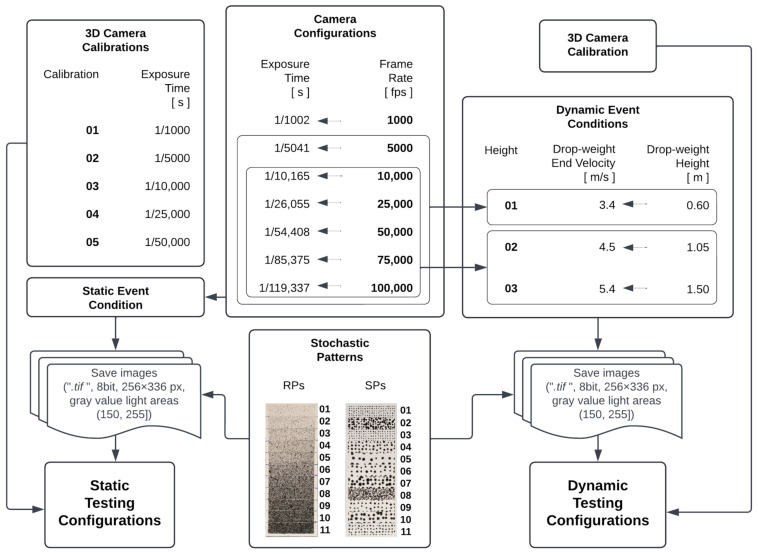
Testing configurations such as camera calibration, exposure time, drop-weight end velocity ($\sqrt{2\times acceleration_{gravity}\times Height}$), and pattern quality for 3D measurements in static and dynamic event conditions.

**Figure 3 sensors-23-02158-f003:**
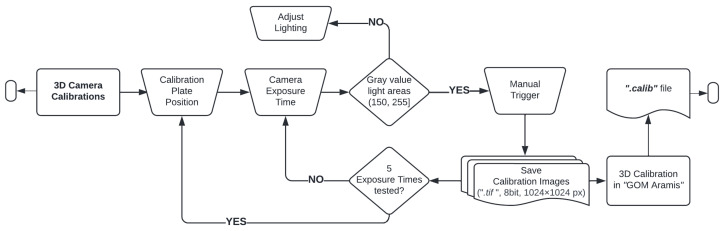
Camera calibration process for determining 3D coordinates in static event condition.

**Figure 4 sensors-23-02158-f004:**
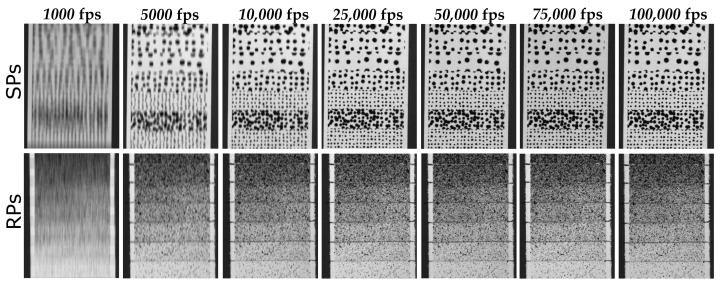
Resulting blurring effect in images recorded at the highest tested velocity with the 7 image frequencies.

**Figure 5 sensors-23-02158-f005:**
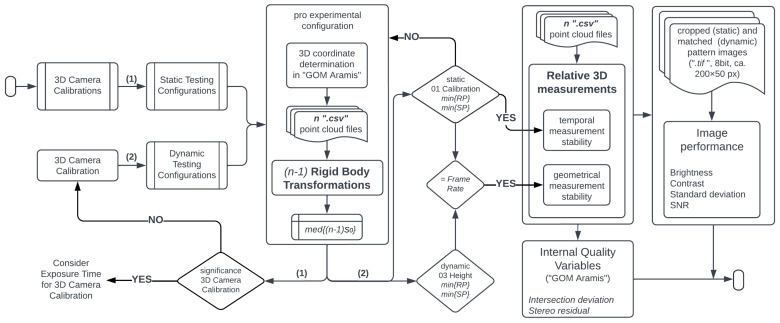
Workflow for the evaluation of the accuracy potential of high frequency measurements.

**Figure 6 sensors-23-02158-f006:**
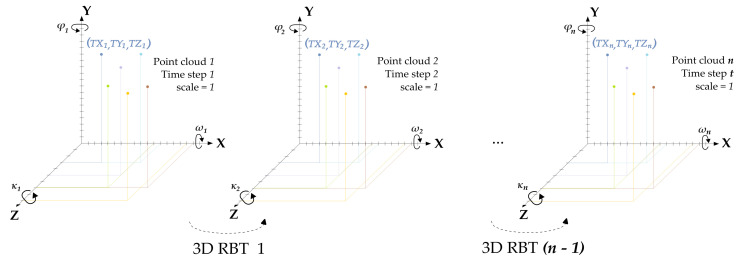
Rigid-body transformations (RBT): The transformation *n* is performed for the point clouds *n* and (n+1).

**Figure 7 sensors-23-02158-f007:**
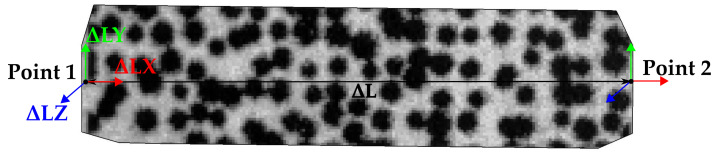
Length variation (ΔL) between two points, and variation per axis direction ΔLX, ΔLY, and ΔLZ.

**Figure 8 sensors-23-02158-f008:**
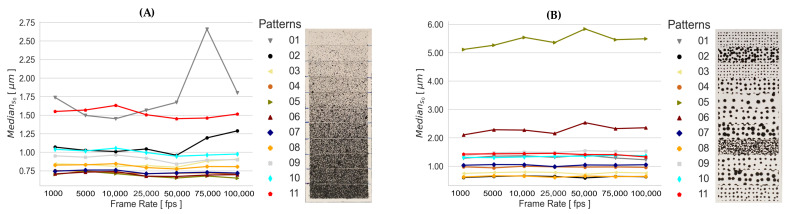
Median $s_{0}$ of the transformations obtained from the static experimental configurations using the calibration parameter set of 04 Calibration, as presented in Figure 2, for (**A**) RPs and (**B**) SPs.

**Figure 9 sensors-23-02158-f009:**
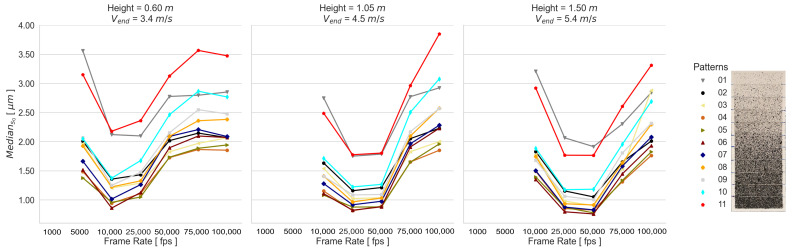
Median $s_{0}$ of the transformations obtained from the dynamic RP experimental configurations.

**Figure 10 sensors-23-02158-f010:**
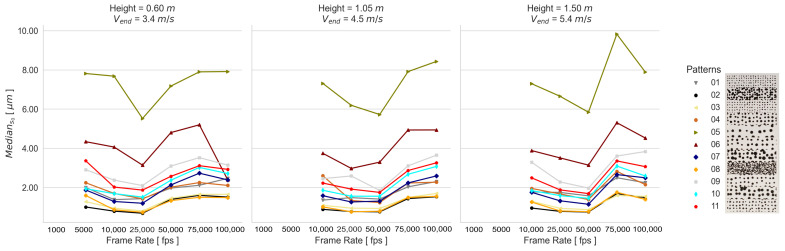
Median $s_{0}$ of the transformations obtained from the dynamic SP experimental configurations.

**Figure 11 sensors-23-02158-f011:**
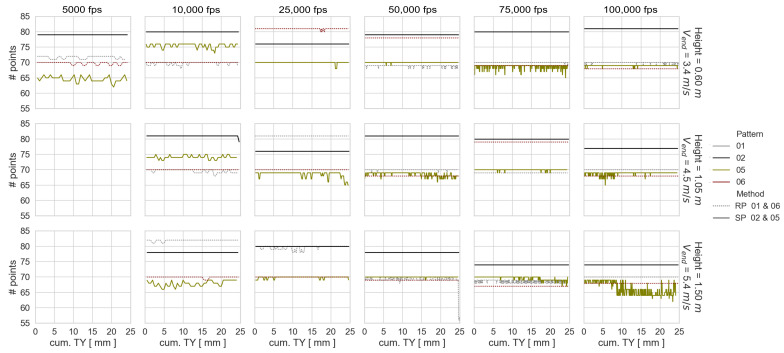
Number of tracked points per time step and vertical cumulative object translation (TY).

**Figure 12 sensors-23-02158-f012:**
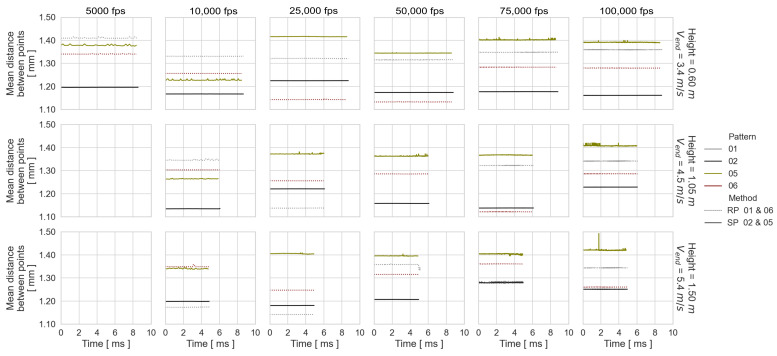
Mean distance between points per time step and object falling recording time.

**Figure 13 sensors-23-02158-f013:**
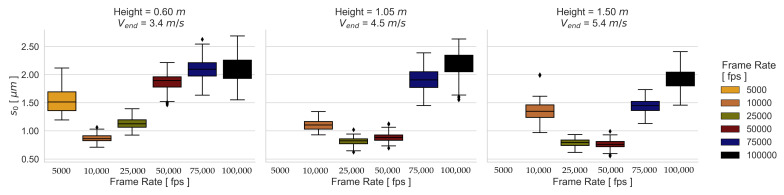
Boxplot of $s_{0}$ errors using RP 06.

**Figure 14 sensors-23-02158-f014:**
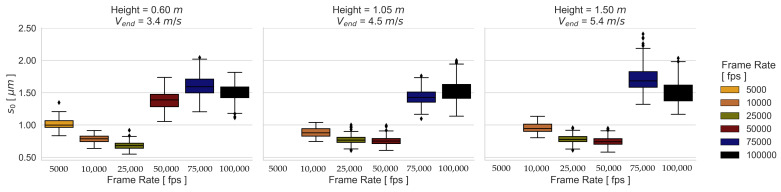
Boxplot of $s_{0}$ errors using SP 02.

**Figure 15 sensors-23-02158-f015:**
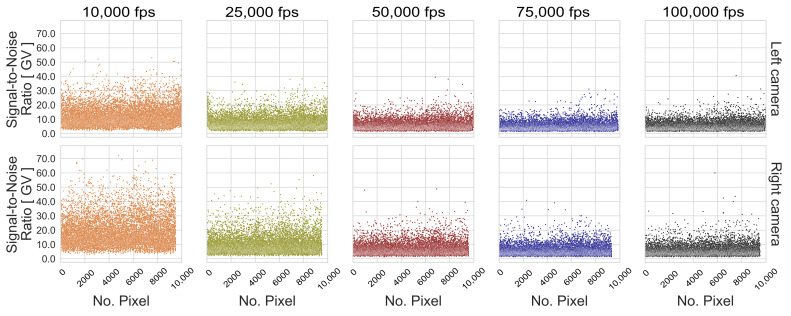
Signal-to-noise ratio per pattern image pixel ($SNR_{P_{px}}$) of RP 06, considering the 5 recorded exposure times at the 03 Height.

**Figure 16 sensors-23-02158-f016:**
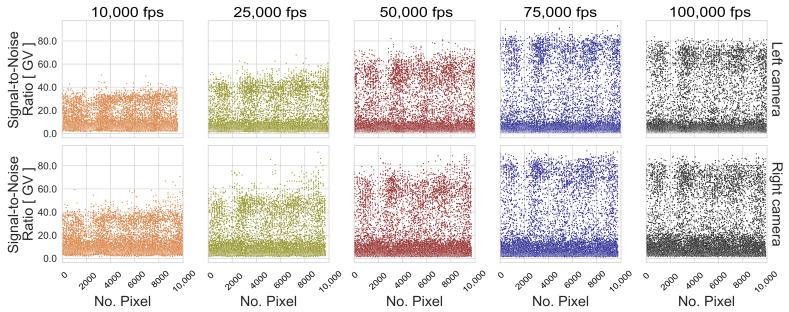
$SNR_{P_{px}}$ of SP 02, considering the 5 recorded exposure times at the 03 Height.

**Figure 17 sensors-23-02158-f017:**
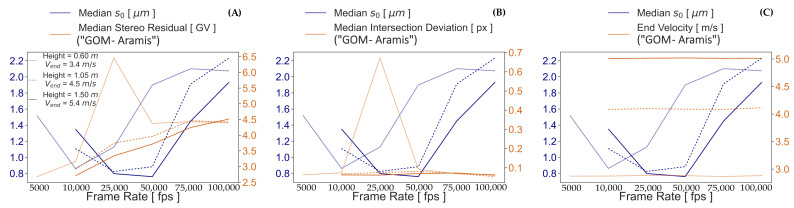
Median $s_{0}$ using the RP 06 vs. (**A**) median *intersection deviation*, (**B**) median *stereo residual*, and (**C**) end velocity.

**Figure 18 sensors-23-02158-f018:**
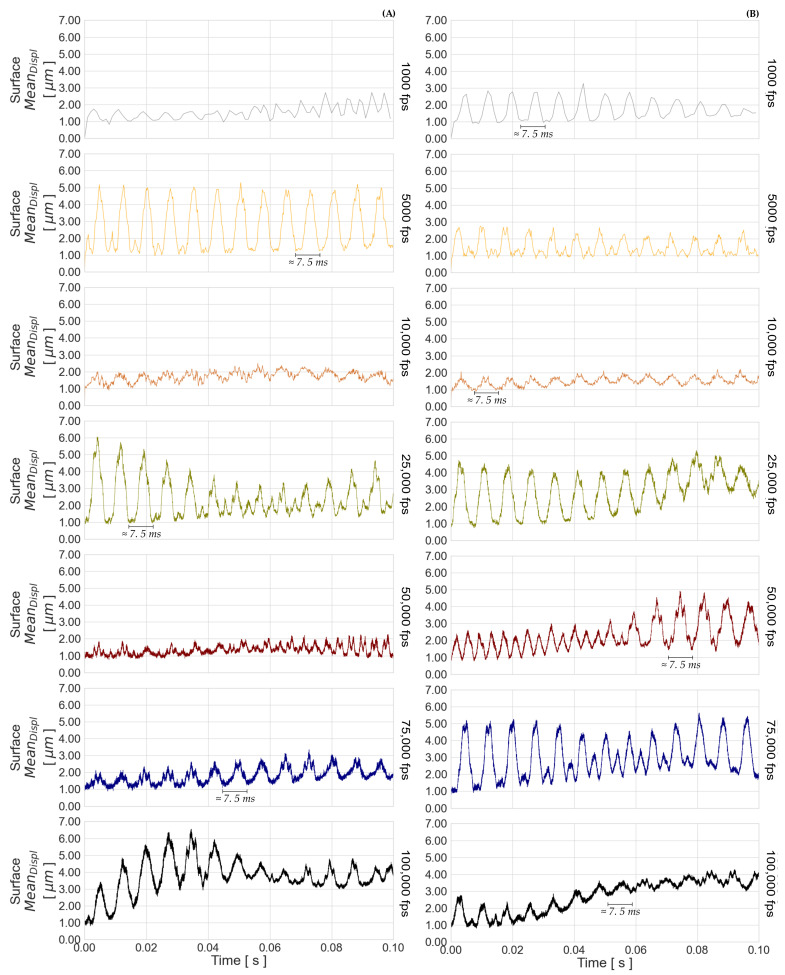
Apparent displacement (DXYZ) of the surface for (**A**) RP 06 and (**B**) SP 02.

**Figure 19 sensors-23-02158-f019:**
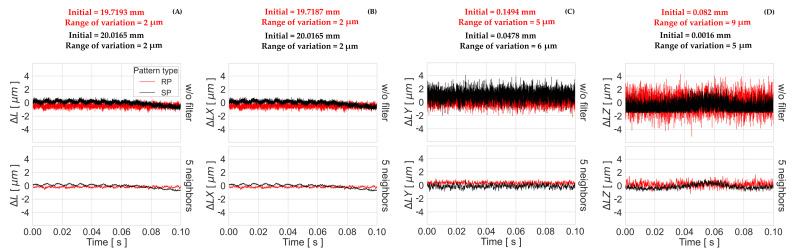
Length variation (**A**) ΔL, (**B**) ΔLX, (**C**) ΔLY, and (**D**) ΔLZ for the best-performing patterns (RP 06 and SP 02) under static conditions using the raw data and a mean filter of 5 neighbors on the time axis.

**Figure 20 sensors-23-02158-f020:**
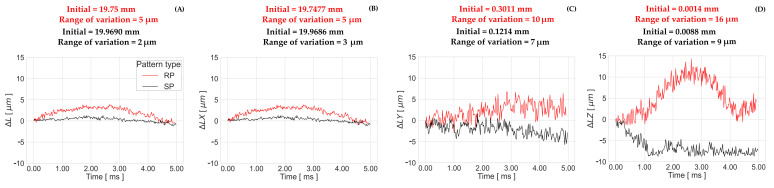
(**A**) ΔL, (**B**) ΔLX, (**C**) ΔLY, and (**D**) ΔLZ for the best-performing patterns (RP 06 and SP 02) under dynamic conditions (03 height, 50000 fps).

**Figure 21 sensors-23-02158-f021:**
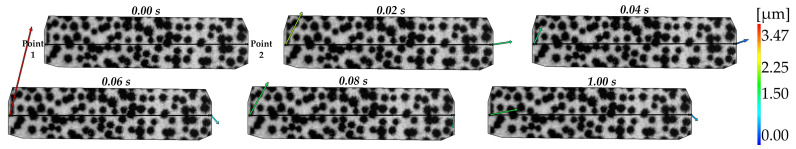
Point vector displacements of static experiment using SP 02 at a frame rate of 50000 fps.

**Table 1 sensors-23-02158-t001:** Main technological specifications of the high-speed camera *“Photron Fastcam SA-X2”*. Data from [17].

Sensor Technology	*Proprietary Design Advanced CMOS*
**Sensor Type**	Monochrome
**Full Sensor resolution**	1024×1024 px at 12,500 fps
**Maximum Frame Rate**	480,000 fps for 128 px × 48/40/8 px
**Sensor Size**	20.48×20.48 mm
**Sensor Diagonal**	28.96 mm
**Pixel** (px) **Size**	20×20 μm
**Fill Factor** (photosensitive area/pixel size)	58%
**Minimum Exposure Time**	Global electronic shutter to 1 μs
**Light Sensitivity**	*ISO* 25,000
**Quantum Efficiency** (% photons converted to photo-electrons)	46% at 630 nm

**Table 2 sensors-23-02158-t002:** Experimental configurations (or data sets) tested under static and dynamic event conditions.

Event Condition	# Camera Calibration	Height	# Frame Rate	# Stochastic Patterns	# Experimental Configurations
Static	5	–	7	11 SP & 11 RP	770
		01	6		132
Dynamic	1	02	5	11 SP & 11 RP	110
		03	5		110

**Table 3 sensors-23-02158-t003:** Median $s_{0}$ in [μ m] for the best- and worst-performing SPs and RPs. Depending on experimental configuration, the lowest $s_{0}$ values are marked in bold.

		5000fps	10,000 fps	25,000 fps	50,000 fps	75,000 fps	100,000 fps
	Static	5.28	5.58	**5.38**	**5.84**	5.48	5.55
**SP 05**	01 Height	7.81	7.68	**5.52**	7.17	7.90	7.91
	02 Height	–	7.31	6.19	**5.72**	7.91	8.43
	03 Height	–	7.29	6.65	**5.84**	9.82	7.89
	Static	0.64	0.66	**0.64**	**0.59**	0.64	0.63
**SP 02**	01 Height	1.00	0.79	**0.68**	1.39	1.59	1.51
	02 Height	–	0.88	0.77	**0.75**	1.43	1.52
	03 Height	–	0.95	0.78	**0.74**	1.68	1.48
	Static	1.50	**1.45**	**1.57**	**1.67**	2.66	1.80
**RP 01**	01 Height	3.56	**2.12**	2.10	2.78	2.80	2.85
	02 Height	–	2.75	**1.75**	1.79	2.77	2.92
	03 Height	–	3.21	2.07	**1.92**	2.30	2.84
	Static	0.73	**0.74**	**0.68**	**0.67**	0.69	0.70
**RP 06**	01 Height	1.52	**0.86**	1.13	1.90	2.10	2.07
	02 Height	–	1.11	**0.82**	0.88	1.91	2.23
	03 Height	–	1.35	0.80	**0.76**	1.45	1.93

**Table 4 sensors-23-02158-t004:** Range variation of apparent displacements (total DXYZ, and for axis direction DX, DY, and DZ) for RP 06, in [μm].

								RP 06
**Frame Rate**	DXYZ	|DXYZ|	DX	|DX|	DY	|DY|	DZ	|DZ|
1000 fps	[0.00, 2.73]	2.73	[ 0.00, 0.91]	0.91	[−0.57, 0.01]	0.58	[−2.60, 2.59]	5.19
5000 fps	[0.00, 5.30]	5.30	[−1.03, 0.58]	1.61	[−0.26, 0.33]	0.59	[−5.26, 2.19]	7.45
10,000 fps	[0.00, 2.50]	2.50	[−1.61, 0.07]	1.68	[−0.12, 1.11]	1.23	[−0.41, 1.81]	2.22
25,000 fps	[0.00, 6.05]	6.05	[−0.83, 1.62]	2.45	[−0.84, 0.25]	1.09	[−2.01, 6.00]	8.01
50,000 fps	[0.00, 2.26]	2.26	[−0.86, 0.33]	1.19	[−0.28, 0.56]	0.84	[−2.17, 2.05]	4.22
75,000 fps	[0.00, 3.35]	3.35	[−0.31, 1.47]	1.78	[−0.41, 0.58]	0.99	[−3.14, 0.77]	3.91
100,000 fps	[0.00, 6.55]	6.55	[−3.91, 0.15]	4.06	[−1.76, 0.70]	2.46	[−3.19, 6.03]	9.22

**Table 5 sensors-23-02158-t005:** Range variation of apparent displacements DXYZ, DX, DY, and DZ for SP 02, in [μm].

								SP 02
**Frame Rate**	DXYZ	|DXYZ|	DX	|DX|	DY	|DY|	DZ	|DZ|
1000 fps	[0.00, 3.27]	3.27	[−0.74, 0.11]	0.85	[−0.37, 0.69]	1.06	[−3.21, 0.61]	3.82
5000 fps	[0.00, 2.75]	2.75	[−0.28, 0.46]	0.74	[−0.24, 0.61]	0.85	[−1.45, 2.71]	4.16
10,000 fps	[0.00, 2.20]	2.20	[−0.21, 1.42]	1.63	[−0.45, 0.36]	0.81	[−0.63, 1.88]	2.51
25,000 fps	[0.00, 5.32]	5.32	[−0.19, 0.96]	1.15	[−0.90, 0.35]	1.25	[−0.70, 5.30]	6.00
50,000 fps	[0.00, 4.92]	4.92	[−0.92, 0.36]	1.28	[−1.67, 0.22]	1.89	[−2.79, 4.74]	7.53
75,000 fps	[0.00, 5.63]	5.63	[−1.31, 0.09]	1.40	[−1.96, 0.25]	2.21	[−5.24, 3.15]	8.39
100,000 fps	[0.00, 4.28]	4.28	[−0.07, 2.75]	2.82	[−2.25, 0.23]	2.48	[−1.51, 3.00]	4.51

## Data Availability

The authors support open scientific exchange to achieve best practices in sharing and archiving research data. Complete experimental data and results are available upon request from the author responsible for the correspondence.

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
