# Peer review of "Analysis of the Accuracy Potential of a Stereo High-Speed Camera System in 3D Measurements in Highly Dynamic Experiments"

_sensors, 2023, doi:10.3390/s23042158_

Round 1
Reviewer 1 Report
The paper is well structured and organized with clearly defined objectives, results and analysis. Methodological aspects are well covered. The results represent a contribution to the academic community
Reviewer 2 Report
Overall, the paper discusses an interesting topic regarding the potential use of a stereo camera system in order to investigate rigid-body transformation. However, I have found several issues that may request further clarity from the authors, as follows:
- Abstract: who will benefit from this study?
- Introduction: The authors need to differentiate a section of general background and existing studies for the sake of clarity.
- Page 4: Table 1 needs references
- In general, the authors use a lot of figures and tables but do not find significant analysis, and only contain very little description. For example, Page 4: the authors need to explain/elaborate further on Figure 2 and Table 2
- Methodology: given the complexity of methods proposed by authors, I highly suggest generating a workflow that describes a procedural step of the study.
- The authors need to elaborate further conclusion part. Limitations and further recommendations are also required.

Reviewer 3 Report
1. For me, the title should not include the initial part, i.e. "Methods for", because the article describes specific studies, not "methods"
2. Reference 8 does not contain a complete description
3. Line 229 is not correct
Round 2
Reviewer 2 Report
Dear Authors
Thank you for responding to my feedback.
One small comment: I guess the general background should be put first before related works in the introduction.
